# Comparison of Hydraulic and Aeration Properties of Peat Substrates Used to Produce Containerized White Spruce Seedlings (1+0) in Forest Nurseries

Simon Boudreault [1], Jean Caron [1], Mohammed S. Lamhamedi [2] and Steeve Pepin [1,*]

[1] Centre de Recherche et d'Innovation sur les Végétaux, Département des Sols et de Génie Agroalimentaire, Pavillon de l'Envirotron, Université Laval, 2480 boul. Hochelaga, Québec, QC G1V 0A6, Canada; sboudreault@activaenviro.ca (S.B.); jean.caron@fsaa.ulaval.ca (J.C.)

[2] Ministère des Ressources Naturelles et des Forêts du Québec, Direction de la Recherche Forestière, 2700 Rue Einstein, Québec, QC G1P 3W8, Canada; mohammed.lamhamedi@mffp.gouv.qc.ca

* Correspondence: steeve.pepin@fsaa.ulaval.ca

**Abstract:** The physical properties of peat substrates from eight tree nurseries were characterized to determine bulk density, air-filled porosity, saturated hydraulic conductivity, pore effectiveness, relative gas diffusivity and chemical properties. There were significant variations among nurseries both in growth of white spruce [*Picea glauca* (Moench) Voss] seedlings (1+0) and substrate properties. Shoot dry mass and root collar diameter were negatively correlated with air-filled porosity and saturated hydraulic conductivity, whereas root dry mass was positively correlated with bulk density. Seedling growth increased with increasing substrate bulk density up to ~0.11 g cm$^{-3}$, above which value conditions may become limiting to plant performance. Our results suggest that there was no growth limitation due to restricted aeration ($D_s D_0^{-1}$ > 0.005 m$^2$ s$^{-1}$ m$^{-2}$ s for all substrates except one) and that over-aeration reduced seedling growth under dry irrigation management.

**Keywords:** air content; bulk density; gas diffusivity; growing media; *Picea glauca*; tree nursery; container; peat moss

## 1. Introduction

Nearly 147 million forest seedlings are produced annually in the 18 forest nurseries (6 government and 12 private) in Quebec, Canada [1]. In 2021, white spruce [*Picea glauca* (Moench) Voss] accounted for 22.6% of the seedlings produced in Quebec, i.e., nearly 30 million seedlings [2]. Before planting, the seedlings are evaluated according to morphophysiological quality criteria and standards established by the Ministry of Natural Resources and Forests (QC, Canada) [3]. Among these criteria, root insufficiency is one of the main causes of plant rejection [4]. In the case of root insufficiency, the plant will be rejected if the root plug partially or completely breaks after extraction from the cavity, shows distinct portions bound by an undamaged root system with more than 5 mm of discontinuity between the portions or if more than 33% of the roots located at the periphery are dead or necrotic [2]. These discarded seedlings obviously have a negative impact on the profitability and long-term viability of nurseries. Various studies have focused on the control and optimization of the main cultural practices, in particular irrigation, fertilization and the model of the container (shape and volume of the cavity, number of seedlings m$^{-2}$) to improve growth and root architecture [5–8]. However, few studies have focused on the evaluation of the physicochemical properties of substrates in relation to the growth of forest seedlings at an operational scale.

Extreme conditions of substrate water content such as excess water and water stress negatively influence plant physiology, root growth and mineral nutrition [6,9–12]. In addition, net photosynthesis decreases with increasing substrate water content when

aeration becomes a limiting factor [13]. Optimizing the physicochemical and aeration properties of the substrate is a critical factor to further improve root growth.

A good artificial growing substrate should have good aeration, high water retention properties, relatively high cation exchange capacity (CEC), be easy to handle and available at low cost [14]. The substrates used in forest nurseries in Quebec are essentially composed of peat and a small proportion of vermiculite and/or perlite. It is common practice in forest nurseries in Quebec to use blond peat with little humification characterized by a high water retention capacity and air content. It has a low pH and a relatively high CEC, two chemical properties ideal for the production and growth of containerized forest seedlings [15]. However, the physicochemical properties of peat can vary depending on its origin and degree of humification [11,16]. In addition, the supply of low-humified peat tends to become difficult with the increase in the duration of peatland exploitation and the scarcity of those exploited near nurseries [17]. The different stages of peat handling, from harvesting to potting, can lead to changes in its physical properties. For example, the type of mixer used to prepare the substrate and a long mixing time can increase the proportion of fine particles [13]. Inappropriate humification and handling of the peat will have the effect of increasing the water retention capacity of the substrate and reducing its air-filled porosity ($\theta_a$), thus increasing the risk of root asphyxiation and waterlogging [11]. Moreover, the height of the saturated zone at the bottom of the root plug after irrigation or rainfall events increases in fine-textured peat substrates, which may further limit gas exchange in the rhizosphere [15].

The physical properties of the substrate can influence the growth of plants [14,18] and forest seedlings [9,13,19,20], in particular air-filled porosity ($\theta_a$), which is largely used as an index of aeration in peat substrates. However, gas exchange between substrates and ambient air also depends on pores' connection in the growing medium; consequently, high $\theta_a$ values are not always indicative of adequate substrate aeration. The relative diffusivity of gases ($D_s\ D_0^{-1}$; estimated from measurements of $\theta_a$, pore tortuosity and saturated hydraulic conductivity) has been shown to be a better parameter than air-filled porosity to investigate how substrate aeration properties affect the development and growth of plant species [21,22]. Our previous study showed that the relative gas diffusivity of peat substrates should not be lower than 0.003 to 0.005 $cm^2\ s\ cm^{-2}\ s^{-1}$ for growth of white spruce seedlings [20]. However, to our knowledge, no study has verified the effect of $D_s\ D_0^{-1}$ in peat–vermiculite potting substrates on the growth of roots and shoots of forest seedlings produced in containers under tunnels during the first growing season in several tree nurseries.

The general objective of this study consists in characterizing the substrates used in the forest nurseries of Quebec using the index of their relative diffusivity of gases and their physicochemical properties in relation to the growth of white spruce seedlings (1+0) during the first growing season under forest nursery conditions. Physical properties were determined directly in the containers to minimize the impact of handling on the fragile structure of the artificial substrates.

## 2. Materials and Methods

### 2.1. Substrate Samples from Forest Nurseries

Peat substrates were collected at the beginning and end of the growing season from 8 out of 18 tree nurseries that produce containerized white spruce seedlings in Quebec (Figure 1). These nurseries were located in different ecological regions (bioclimatic domains), between 46°04′–48°49′ N lat. and 65°51′–74°38′ W long. (Figure 1). Accordingly, the number of growing degree-day units (mean air temperature above 5 °C) during the vegetative season varied between 1250 and 2000 among the different nurseries. For confidentiality purposes, we assigned a number from 1 to 8 to each forest nursery included in the study (see Section 3.2).

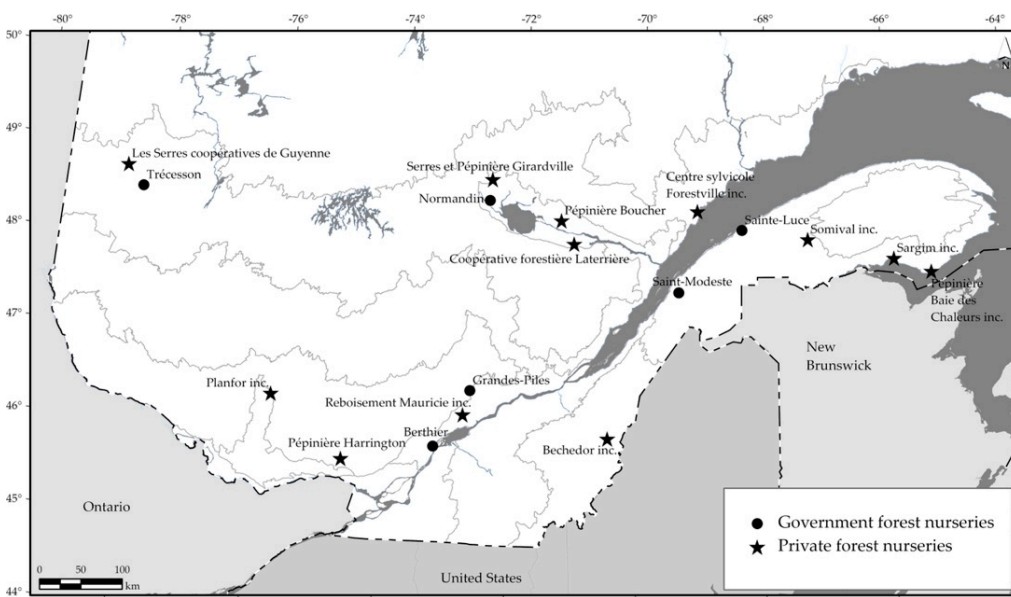

**Figure 1.** Locations of private and government forest nurseries in the province of Québec (adapted from Lamhamedi et al. [23]).

### 2.2. Production of White Spruce Seedlings (1+0)

White spruce seeds were sown into containers (model IPL 25-310: 25 cavities per container, 310 cm$^3$ per cavity, each 12 cm in height with top and base diameters of 6.5 cm and 5.2 cm; IPL$^®$, Saint-Damien-de-Buckland, QC, Canada) between 7 and 20 May 2008, depending on the ecological region. Containers were filled with substrates composed of sphagnum peat moss of various grades (specific to each nursery) provided by Quebec and New Brunswick peat suppliers. Horticultural grade 3 or 4 vermiculite was added to all substrates in proportions that varied among nurseries from 9 to 40% ($v\,v^{-1}$); only the substrate of nursery 8 included perlite (9%, $v\,v^{-1}$). White spruce seedlings were cultivated under unheated polyethylene tunnels, with irrigation and fertilization management practices tailored to the operational conditions of each tree nursery. The amount of nitrogen (N), phosphorous (P), potassium (K) and oligo-nutrients that seedlings received during the growing season (see Table 1) were based on the fertility of peat substrates and seedlings' developmental stage and nutritional needs in order to meet the required growth standards for white spruce (1+0) stock [24,25]. The irrigation was adjusted by nursery managers according to seedling size; the volumetric water content ($\theta_v$) of substrates was determined using time domain reflectometry [6,26] or gravimetric methods [27]. Typically, irrigation management under tunnel resulted in substrate water contents of about 0.35 to 0.45 cm$^3$ cm$^{-3}$.

**Table 1.** Composition, chemical properties, and initial fertility of substrates after potting ($n$ = 5), and quantities of nitrogen (N), phosphorous (P), and potassium (K) applied per seedling of white spruce (1+0) during the growing season in sampled forest nurseries.

| No. [1] | Peat | Vermi [2] | Perlite | pH | EC [3] (mS m$^{-1}$) | CEC$_{eff}$ [4] (meq L$^{-1}$) | Initial Fertility (µg 100 cm$^{-3}$) | | | | | | Applied Fertilizer (mg Plant$^{-1}$) | | |
| | | (%, $v\,v^{-1}$) | | | | | N_NH$_4$ | N_NO$_{2,3}$ | P | K | Ca | Mg | N$_{total}$ | P | K |
|---|---|---|---|---|---|---|---|---|---|---|---|---|---|---|---|
| 1 | 75 | 25 | 0 | 4.20 | 116.7 | 26.3 | 14 | 2 | 16 | 57 | 4 | 27 | 86 | – | – |
| 2 | 75 | 25 | 0 | 3.81 | 209.0 | 22.4 | 13 | 6 | 18 | 60 | 7 | 35 | 34 | 17 | 21 |
| 3 | 80 | 20 | 0 | 4.10 | 126.0 | 21.6 | 32 | 5 | 51 | 52 | 5 | 21 | 106 | 24 | 38 |
| 4 | 86 | 15 | 0 | 4.14 | 119.7 | 26.3 | 14 | 19 | 17 | 37 | 4 | 19 | 83 | 19 | 30 |
| 5 | 75 | 25 | 0 | 4.20 | 166.0 | 32.2 | 28 | 2 | 90 | 90 | 12 | 39 | 82 | 27 | 49 |
| 6 | 80 | 20 | 0 | 4.17 | 213.3 | 33.9 | 17 | 10 | 53 | 157 | 15 | 5 | 51 | 12 | 26 |
| 7 | 60 | 40 | 0 | 4.00 | 161.3 | 24.4 | 18 | 7 | 7 | 40 | 4 | 26 | 51 | 9 | 21 |
| 8 | 82 | 9 | 9 | 3.99 | 143.6 | 25.0 | 29 | 9 | 8 | 19 | 5 | 29 | 48 | 13 | 31 |

[1] Forest nursery number. [2] Vermi = vermiculite. [3] EC = electrical conductivity. [4] CEC$_{eff}$ = effective cation exchange capacity.

### 2.3. Physical Properties of Peat Substrates

One week after potting and prior to seeding, five containers were randomly selected in each tree nursery and brought back to the laboratory to determine the physical properties of the substrates in situ (i.e., on root plugs in containers). Bulk density ($\rho_b$) was evaluated for three individual cavities per container by measuring the volume and dry weight (oven-dried at 105 °C for 24 h) of each root plug. Total porosity ($\theta_t$) was measured on the same three samples by determining the organic matter content after calcination at 550 °C [28]. Particle size distribution was measured for 650 mL air-dried composite substrate samples ($n$ = 3 root plugs) sieved mechanically (shaker model RX-29, W. S. Tylers®, Mentor, OH, USA) for 3 min on a nest of sieves (0.106, 0.25, 0.5, 1, 2, 4 and 8 mm stacked over a pan). The mean weight diameter (MWD) of particles was calculated as described by Kemper and Roseneau [29]:

$$\text{MWD} = \sum_{i=1}^{n} x_i f_i \tag{1}$$

where $x_i$ is the weight fraction retained on the $i$th sieve divided by the total substrate sample mass, $f_i$ is the average particle size on the $i$th sieve, and $n$ is the number of size classes, here equal to 8.

Containers with remaining root plugs were slowly saturated by gradually increasing the water level at a rate of 0.5 cm h$^{-1}$ for 24 h; substrates were then drained freely for 2 h to determine the volumetric water content at container capacity ($\theta_c$, which corresponds to a matric potential of −0.6 kPa, i.e., at mid-height of the IPL 25–310 container). Substrate water content was measured using a 10.5 cm long three-rod (0.2 cm in diameter, spaced at 1 cm apart) TDR probe inserted vertically within a cavity and a cable tester (model 1502B, Tektronix Inc., Beaverton, OR, USA) (Figure 2). For all $\theta_v$ determinations, dielectric constants ($K_a$) between 5 and 50.8 were converted to $\theta_v$ using the $K_a$–$\theta_v$ equation established by Paquet et al. [30], which was found appropriate for these substrates [20]. Values of $K_a$ between 50.8 and 81 were linearly interpolated from $\theta_v$ at $K_a$ = 50.8 to $\theta_v$ = 1.0 cm$^3$ cm$^{-3}$ at $K_a$ = 81. The air-filled porosity after drainage ($\theta_a$) was calculated as

$$\theta_a = \theta_t - \theta_c \tag{2}$$

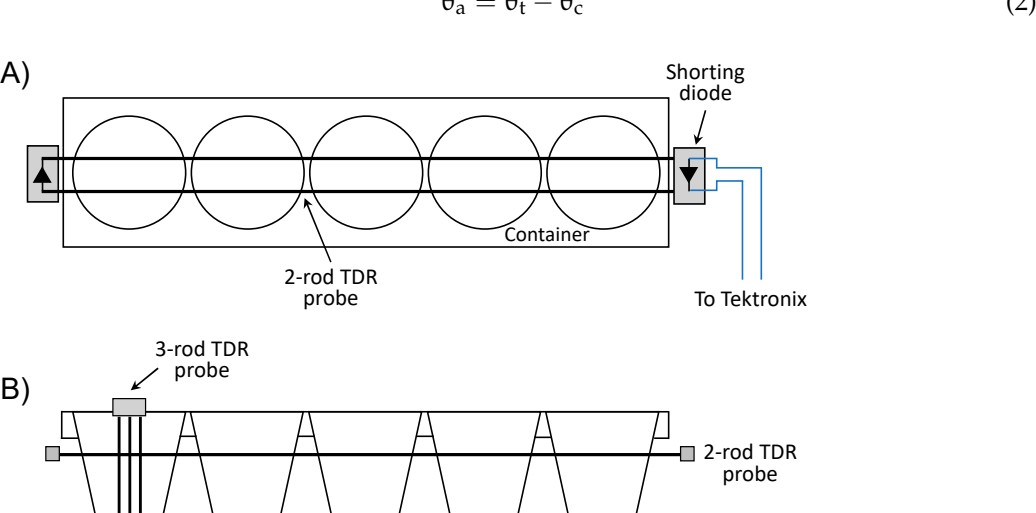

**Figure 2.** Schematic of the linear five-cavity container (aerial (**A**) and lateral (**B**) views) including a 10.5 cm long three-rod TDR probe inserted vertically within a cavity and a 36 cm long two-rod TDR probe with shorting diodes inserted horizontally across the five cavities at a height of 8 cm from the bottom of the container.

Five substrate root plugs were carefully extracted from each nursery container and placed in a linear five-cavity container, with the lower end of the 310 cm$^3$ cavities covered with mosquito netting. The five-cavity containers were slowly saturated as described above and then placed on a tension table under a plastic cover. Substrate moisture content was

determined using a 36 cm long two-rod (1.5 cm spacing) TDR probe with shorting diodes at the beginning and end of the probe, inserted horizontally into the container across the five cavities at a height of 8 cm from the bottom of the cavities [26]. A correction factor was used for the calculation of $K_a$ to account for propagation time that was not related to the substrate within the five cavities [26].

To determine the matric potential at which air first enters the substrate when going from saturated to unsaturated conditions ($\psi_a$, the point of air entry; Nemati et al. [31]), the five-cavity containers were rewetted from the bottom up to full saturation overnight and then placed on a tension table with a water potential of 0.6 kPa (i.e., a water head of 6 cm above the surface of the tension table). The matric potential ($\psi$) at mid-container height was reduced from 0 to $-0.8$ kPa by decreasing the water level in 0.2 kPa steps and the substrate $\theta_v$ was measured via TDR at each $\psi$ step following a 2 h equilibrium period. From these $\psi$–$\theta_v$ curves, $\psi_a$ was estimated visually and corresponded to the matric potential when a significant drop in $\theta_v$ was observed. If air entry had not occurred at $-0.8$ kPa, substrate $\psi$ was further reduced from $-0.8$ to $-1.2$ kPa. For these measurements, matric potentials were evaluated from the height of the water level above the surface of the tension table, from which 0.125 kPa was subtracted to account for the zone of influence of the TDR probe (i.e., the distance required to detect a change in $\theta_v$; see Caron et al. [32] and Boudreault et al. [20]). Afterward, substrates were covered by a plastic sheet to prevent evaporation and their water release curve was established by measuring $\theta_v$ at matric potentials of $-1.4$, $-1.6$, $-1.8$, $-3$, $-5$ and $-10$ kPa. The available (AW) and easily available (EAW) water content in the substrates were estimated from $\theta_c$ ($\psi = -0.6$ kPa) and values of $\theta_v$ at $\psi = -10$ kPa and $-5$ kPa, respectively [33].

$$EAW = \theta_c - \theta_{-5kPa} \tag{3}$$

$$AW = \theta_c - \theta_{-10kPa} \tag{4}$$

Saturated hydraulic conductivity ($K_s$) was determined on three individual cavities per container per nursery using an infiltrometer that provided a constant flow of water at the surface of the sample [28]. Substrates were saturated as described above; once steady-state conditions of water flow through the substrate were achieved, the pressure gradient and the flow of water were measured and $K_s$ estimated using Darcy's law. A correction factor of 1.22 was used to account for the slightly conical shape of the cavity and narrowing of the wall near the bottom of the cavity (see Figure 1 in Boudreault et al. [20]). The aeration properties of the substrates were then determined using measured values of $\psi_a$ and $K_s$ and data from the water-release curve based on the multiple-point method described by Allaire et al. [18] and Caron and Nkongolo [22]. Briefly, this method consists in determining the parameters of the water desorption curve using a nonlinear, five-parameter function adapted to artificial substrates [34]:

$$\theta = \theta_r + \frac{\theta - \theta_r}{\left[1 + (a\psi)^n\right]^b} \tag{5}$$

where $\theta$ is the substrate volumetric water content (cm$^3$ cm$^{-3}$), $\psi$ is the matric potential (kPa), $\theta_r$ is the residual water content (cm$^3$ cm$^{-3}$) and a, b and n are empirical parameters determined by minimizing the error between estimates of Equation (5) and points of the water release curve using Mathcad 7. Once established, the function was used with $K_s$ values to determine the coefficient of pore effectiveness ($\gamma$) following

$$\gamma = \left\{ \frac{0.00028\rho g}{\eta K_s} \int_{\theta_r}^{\theta_{\psi_a}} a^2 \left[ \left( \frac{\theta_s - \theta_r}{\theta - \theta_r} \right)^{-\frac{1}{b}} - 1 \right]^{-\frac{2}{n}} d\theta \right\}^{-1} \tag{6}$$

where $\theta_{\psi a}$ is the water content at air entry value ($cm^3$ $cm^{-3}$), $\theta_s$ is the water content at saturation ($cm^3$ $cm^{-3}$), $\rho$ is water density (g $cm^{-3}$), g is the gravitational acceleration (m $s^{-2}$) and $\eta$ is the viscosity of water (0.001 Pa s at 20°C). Because g represents the tortuosity, constriction and continuity of pores in the substrate, $D_s$ $D_0^{-1}$ can be estimated using the equation from King and Smith [35], generalized for artificial media:

$$\frac{D_s}{D_0} = \gamma \, \theta_a \tag{7}$$

Equation (7) provides an estimate of relative gas diffusivity at an average $\psi$ of $-0.6$ kPa found in the 25–310 nursery containers when substrates are at container capacity.

### 2.4. Chemical Properties of Peat Substrates

Mineral nutrients, effective cation exchange capacity ($CEC_{eff}$), pH and electrical conductivity (EC) were determined for substrates sampled from the eight tree nurseries prior to seeding. The substrate from three cavities per container was air-dried in the laboratory and combined to form a composite sample ($n = 5$ per nursery). Major cations were extracted from each composite sample with a 1 mol $L^{-1}$ solution of $NH_4Cl$ and CECeff was calculated as the sum of exchangeable bases plus actual acidity [36]. The nutrient (cation) composition of each extract was then measured using plasma atomic emission spectrometry (Model ICAP 9000, Thermo Instruments, Franklin, MA, USA) [36] The concentrations of $NH_4$–N, $NO_2$–N, $NO_3$–N, P, K, Ca and Mg in the substrates were measured in saturated aqueous extracts [36,37]. After extraction, the mineral N concentration was determined by colorimetry using a continuous flow spectrophotometer (QuickChem 8000, Lachat Instruments, Loveland, CO, USA), while P, K, Ca and Mg concentrations were determined by atomic emission spectrometry with plasma (ICAP 61E, Thermo Instruments, Franklin, MA, USA) [36,37]. Substrate pH and EC were measured directly on the aqueous extracts without reporting the results on a dry basis [38]. Mineral nutrient analyses of substrates were determined at the organic and inorganic chemistry laboratory of the Quebec Forest Research Branch.

### 2.5. Growth and Nutrition of White Spruce Seedlings (1+0) and Physical Properties of Substrates after One Growing Season

In spring 2009, three containers were randomly selected prior to bud burst in each of the eight tree nurseries to determine the growth parameters and nutrient content of white spruce seedlings (1+0). Seedling height and root collar diameter were measured in the laboratory on ten seedlings randomly selected per container ($n = 30$ per nursery). Seedlings were then severed at the root collar and their root system was carefully washed by hand. The root and shoot dry masses of these seedlings were measured after oven-drying at 70 °C for 48 h. Nutrient concentrations in root and shoot tissues were determined for three composite samples per tree nursery, each consisting of the ten seedlings per container used to determine growth parameters. After tissue grinding and acid digestion, P, K, Ca and Mg concentrations were measured by atomic emission spectrometry with plasma (ICAP 61E, Thermo Instruments), whereas N concentration was determined using the Kjeldahl method (flow injection colorimetry; QuickChem 8000, Lachat Instruments). All mineral nutrient analyses were performed at the organic and inorganic chemistry laboratory of the Quebec Forest Research Branch. Additionally, the bulk density and saturated hydraulic conductivity of substrates collected in spring 2009 were each measured on three different cavities per container. Volumetric water content at saturation (i.e., total porosity, $\theta_t$) and container capacity ($\theta_c$) were also determined to estimate air-filled porosity after drainage ($\theta_a$) using Equation (2).

### 2.6. Statistical Analyses

An analysis of variance followed by a protected LSD test were used to compare substrate physical properties, seedling growth parameters and nutrient concentrations

among tree nurseries (with significance determined at $p < 0.05$). Pearson's correlation coefficients and linear regressions were computed to examine the relationships between growth parameters of white spruce and the physical properties ($\rho_b$, $\theta_a$, EAW, AW, $K_s$, MWD, $\gamma$ and $D_s D_0^{-1}$) of substrates after potting. A similar analysis was performed with values of $\rho_b$, $\theta_a$ and $K_s$ obtained after one year of growth (i.e., on substrates sampled in spring 2009). All statistical analyses were performed using SAS 9.1.3 (SAS Institute Inc., Cary, NC, USA).

## 3. Results

### 3.1. Physical Properties of Forest Nursery Substrates

The mean weight diameter (MWD) of particles varied between 1.17 and 3.06 mm after potting, with a mean ($\pm$SD) value of $1.92 \pm 0.57$ mm across all nurseries (Figure 3). Whereas the fraction of particles with a diameter of 1 to 4 mm varied from 25% in the substrate from nursery 1 to 52% in those from nurseries 5 and 6, there was a five-fold increase in the fraction of fine particles (<0.5 mm) in substrates from nursery 6 compared to those from nursery 1 (Figure 3). A greater water retention was observed in the substrates from nursery 1 and 5 over most of the measured range of matric potentials, while substrates from nurseries 2, 3, 4, 6, 7 and 8 had similar water release curves (Figure 4).

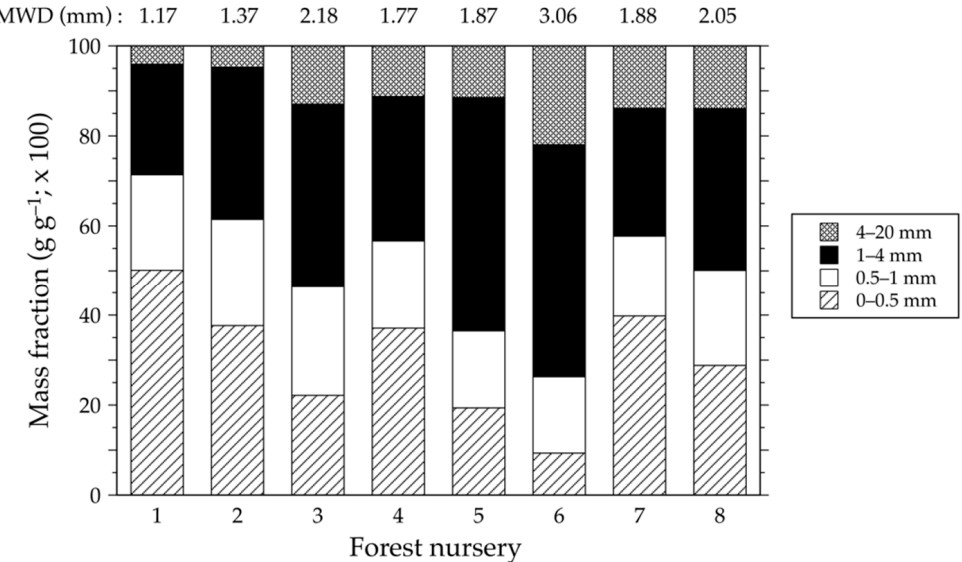

**Figure 3.** Mean weight diameter (MWD; above each stack column) and the proportion of peat particles with a diameter < 0.5 mm and between 0.5 and 1 mm, 1 and 4 mm and 4 and 20 mm in nursery substrates after potting.

The average saturated hydraulic conductivity of substrates after potting was $0.06 \pm 0.01$ cm s$^{-1}$, with values ranging from 0.02 (nursery 1) to 0.11 cm s$^{-1}$ (nursery 6) (Table 2). Substrate bulk density, which varied between 0.076 and 0.105 g cm$^{-3}$, differed significantly ($p < 0.001$) among forest nurseries, those of nursery 1 and 5 having the highest $\rho_b$ values (Table 2). There was little variation in total porosity of the substrates sampled after potting ($0.944 \pm 0.001$ cm$^3$ cm$^{-3}$), yet the air-filled porosity ($\theta_a$) was three times higher in substrates from nurseries 3 and 7 (>0.14 cm$^3$ cm$^{-3}$) compared to that of nursery 1 (0.04 cm$^3$ cm$^{-3}$) (Table 2). While no significant difference was observed in the easily available water (EAW) content of substrates ($p = 0.09$), values of AW differed significantly among nursery substrates ($p < 0.001$). EAW and AW averaged over all substrates after potting were $0.31 \pm 0.04$ cm$^3$ cm$^{-3}$ and $0.42 \pm 0.02$ cm$^3$ cm$^{-3}$, respectively. There was no significant difference in pore effectiveness coefficient ($\gamma$) or relative gas diffusivity ($D_s D_0^{-1}$) among the eight substrates (Table 2). Values of $\gamma$ changed relatively little across the different substrates, ranging from 6.6% in nursery 2 to 11.8% in nursery 6. The substrate from nursery 1 had the

lowest $D_s D_0^{-1}$ value (0.0041 m$^2$ s$^{-1}$ m$^{-2}$ s), whereas the highest relative gas diffusivity was found in the substrate from nursery 3 (0.0171 m$^2$ s$^{-1}$ m$^{-2}$ s), in accordance with its high air-filled porosity.

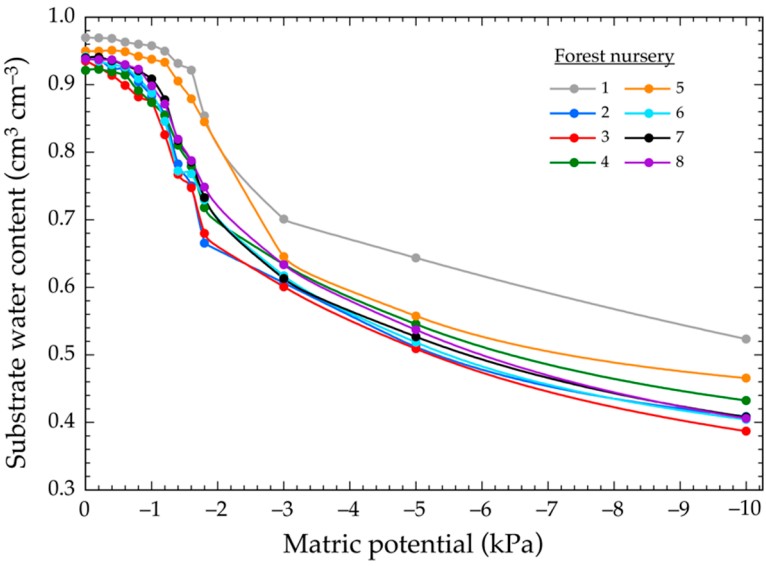

**Figure 4.** Water release curves of substrates from sampled forest nurseries after potting.

**Table 2.** Physical properties of substrates from eight forest nurseries after potting.

| No. [1] | $K_s$ | $\rho_b$ | $\theta_t$ | $\theta_a$ | EAW | AW | $\gamma$ | $D_s D_0^{-1}$ |
|---|---|---|---|---|---|---|---|---|
| | (cm s$^{-1}$) | (g cm$^{-3}$) | (cm$^3$ cm$^{-3}$) | (cm$^3$ cm$^{-3}$) | (cm$^3$ cm$^{-3}$) | (cm$^3$ cm$^{-3}$) | (m m$^{-1}$) | (m$^2$ s$^{-1}$ m$^{-2}$ s) |
| 1 | 0.02 e | 0.095 ab | 0.943 bc | 0.04 d | 0.26 a | 0.38 bc | 0.093 a | 0.0041 a |
| 2 | 0.04 d | 0.089 c | 0.945 bc | 0.11 bcd | 0.29 a | 0.44 ab | 0.066 a | 0.0070 a |
| 3 | 0.09 bc | 0.076 d | 0.954 a | 0.17 a | 0.28 a | 0.40 a | 0.096 a | 0.0171 a |
| 4 | 0.08 bc | 0.094 ab | 0.943 bc | 0.09 c | 0.34 a | 0.41 a | 0.108 a | 0.0088 a |
| 5 | 0.03 e | 0.105 a | 0.938 d | 0.11 bc | 0.26 a | 0.39 c | 0.105 a | 0.0119 a |
| 6 | 0.11 a | 0.099 a | 0.942 bc | 0.08 cd | 0.30 a | 0.46 a | 0.118 a | 0.0091 a |
| 7 | 0.05 d | 0.088 c | 0.945 b | 0.14 ab | 0.28 a | 0.40 a | 0.075 a | 0.0104 a |
| 8 | 0.05 cd | 0.092 bc | 0.944 bc | 0.08 cd | 0.33 a | 0.46 a | 0.081 a | 0.0061 a |
| Mean | 0.06 | 0.091 | 0.944 | 0.11 | 0.31 | 0.42 | 0.087 | 0.0090 |
| SD | 0.01 | 0.002 | 0.001 | 0.02 | 0.04 | 0.02 | 0.020 | 0.0100 |

[1] No. = forest nursery number, $K_s$ = saturated hydraulic conductivity, $\rho_b$ = bulk density, $\theta_t$ = total porosity, $\theta_a$ = air-filled porosity at container capacity, EAW = easily available water content, AW = available water content, $\gamma$ = pore effectiveness coefficient and $D_s D_0^{-1}$ = relative gas diffusivity. SD = standard deviation. Means followed by the same letter are not significantly different at $p = 0.05$ according to a protected LSD test.

### 3.2. Growth of White Spruce Seedlings (1+0)

The provenance of white spruce seedlings had a significant effect on measured growth parameters (Table 3). For instance, there was a 78% difference in shoot dry mass of seedlings between nursery 1 (1244 mg) and nursery 8 (277 mg), while root dry mass differed by 68% among sampled forest nurseries (Table 3). Average seedling height at the end of the growing season varied between 6.4 and 12.9 cm, whereas mean root collar diameter ranged from 1.36 to 3.39 mm. Furthermore, nutrient concentrations in the shoot tissues of white spruce seedlings were significantly different among nursery substrates ($p < 0.05$; Table 3).

### 3.3. Correlations between Seedling Growth Parameters and Substrate Physicochemical Properties

There were no correlations between morphological variables (height, root collar diameter, shoot dry mass, root dry mass) of white spruce seedlings and the 10 physical properties of substrates from eight forest nurseries after potting (Table 4). However, significant correlations were found between growth parameters and substrate physical properties measured at the end of the growing season ($K_{s\_1}$, $\rho_{b\_1}$ and $\theta_{a\_1}$). A significant linear relationship was

observed between saturated hydraulic conductivity ($K_{s\_1}$) and shoot dry weight ($r = -0.53$, $p = 0.008$), root collar diameter ($r = -0.42$, $p = 0.039$) and height/diameter ratio ($r = 0.41$, $p = 0.042$). Air-filled porosity ($\theta_{a\_1}$) was significantly negatively correlated with shoot dry weight ($r = -0.53$, $p = 0.008$; Figure 5) and collar diameter ($r = -0.51$, $p = 0.01$). Additionally, there was a significant positive correlation between the bulk density of substrates ($\rho_{b\_1}$) and root dry weight ($r = 0.63$, $p < 0.001$; Table 4 and Figure 6). Morphological variables were also significantly correlated with the quantity of nitrogen ($N_{total}$), phosphorous (P) and potassium (K) applied per seedling. A positive linear relation was observed between the dose of applied P and shoot dry weight ($r = 0.84$, $p = 0.019$) as well as seedling height ($r = 0.82$, $p = 0.021$). Root collar diameter was correlated with the doses of applied N ($r = 0.80$, $p = 0.030$) and P ($r = 0.86$, $p = 0.011$), whereas the H/D ratio was negatively related to applied N ($r = -0.81$, $p = 0.026$).

**Table 3.** Morphological variables of white spruce seedlings (1+0) and nutrient concentration of N, P, K, Ca and Mg in shoot tissues ($n = 3$).

| No. [1] | Shoot DW (mg) | Root DW (mg) | H (cm) | D (mm) | H/D (cm mm$^{-1}$) | N (g kg$^{-1}$) | P (g kg$^{-1}$) | K (g kg$^{-1}$) | Ca (g kg$^{-1}$) | Mg (g kg$^{-1}$) |
|---|---|---|---|---|---|---|---|---|---|---|
| 1 | 1244 a | 520 a | 9.3 b | 2.9 b | 3.2 e,f | 26.8 a,b | 3.1 b,c,d | 5.5 d,e | 1.4 c | 1.3 d |
| 2 | 1190 a,b | 552 a | 10.0 b | 3.4 a | 3.0 f | 24.1 b,c | 3.2 b | 6.7 c,d | 1.9 a,b | 1.8 a |
| 3 | 997 b,c | 370 b | 9.9 b | 2.7 b | 3.7 d,e | 23.1 b,c,d | 2.6 c,d | 5.4 e | 1.7 b | 1.5 b,c,d |
| 4 | 896 c,d | 355 b | 9.9 b | 2.4 c | 4.2 c | 23.6 b,c,d | 2.6 c,d | 8.1 b | 1.7 b | 1.7 a,b |
| 5 | 812 c,d | 371 b | 12.9 a | 2.4 c | 5.3 a | 21.3 c,d | 3.1 b,c | 18.6 a | 2.1 a | 1.7 a,b,c |
| 6 | 734 d,e | 506 a | 8.2 c | 2.3 c | 3.6 d,e | 19.9 d | 2.6 d | 6.4 b,c | 1.3 c | 1.4 c,d |
| 7 | 511 e,f | 274 b,c | 6.9 d | 1.8 d | 3.8 c,d | 30.7 a | 4.2 a | 7.2 c,d,e | 2.1 a | 1.9 a |
| 8 | 277 f | 177 c | 6.4 d | 1.4 e | 4.7 b | 25.1 b,c | 3.4 b | 6.2 c,d,e | 1.6 b | 1.5 b,c,d |

[1] No. = forest nursery number, DW = dry weight, H = seedling height, D = stem diameter, H/D = height–diameter ratio. Forest nurseries were sorted from 1 to 8 in tables and figures based on shoot dry weight of white spruce seedlings (1+0) at the end of the growing season. Means followed by the same letter are not significantly different at $p = 0.05$ according to a protected LSD test.

**Table 4.** Pearson correlation coefficients [1] between morphological variables and physical and chemical properties of forest nursery substrates after potting, quantities of nitrogen ($N_{total}$), phosphorous (P) and potassium (K) applied per seedling of white spruce (1+0) during the growing season and physical properties determined at the end of the growing season.

| | Shoot DW [2] | Root DW | H | D | H/D |
|---|---|---|---|---|---|
| Physical properties after potting | | | | | |
| $K_s$ | 0.02 | 0.28 | −0.18 | 0.17 | −0.41 |
| $\rho_b$ | −0.03 | 0.28 | 0.31 | 0.00 | 0.34 |
| $\theta_t$ | 0.09 | −0.19 | −0.24 | 0.06 | −0.35 |
| $\theta_a$ | −0.18 | −0.39 | 0.11 | −0.10 | 0.17 |
| MWD | −0.62 | −0.36 | 0.07 | −0.58 | 0.77 |
| <0.5 mm | 0.30 | 0.06 | −0.29 | 0.17 | −0.50 |
| 1_4 mm | −0.24 | 0.06 | 0.10 | −0.16 | 0.24 |
| EAW | −0.25 | 0.11 | −0.44 | −0.08 | −0.27 |
| AW | −0.45 | −0.08 | −0.50 | −0.30 | −0.05 |
| $\gamma$ | 0.34 | 0.66 | 0.35 | 0.47 | −0.21 |
| $D_s D_0^{-1}$ | 0.02 | −0.10 | 0.33 | 0.12 | 0.12 |
| Chemical properties after potting | | | | | |
| pH | −0.11 | −0.07 | −0.30 | −0.32 | −0.06 |
| CE | −0.30 | −0.03 | −0.39 | −0.10 | −0.23 |
| $CEC_{eff}$ | −0.41 | −0.08 | −0.47 | −0.45 | −0.03 |
| $NH_4$ | −0.13 | −0.06 | 0.32 | −0.09 | 0.37 |
| $NO_{2-3}$ | −0.20 | −0.44 | 0.14 | −0.23 | 0.40 |
| $CEC_{eff}$ | −0.41 | −0.08 | −0.47 | −0.45 | −0.03 |
| $NH_4$ | −0.13 | −0.06 | 0.32 | −0.09 | 0.37 |
| $NO_{2-3}$ | −0.20 | −0.44 | 0.14 | −0.23 | 0.40 |
| Ca | −0.32 | 0.02 | −0.43 | −0.23 | −0.24 |
| Mg | 0.32 | 0.61 | 0.30 | 0.41 | −0.13 |

**Table 4.** *Cont.*

| | Shoot DW [2] | Root DW | H | D | H/D |
|---|---|---|---|---|---|
| Applied fertilizer | | | | | |
| $N_{total}$ | 0.72 | 0.63 | 0.52 | **0.80 *** | **−0.81 *** |
| P | **0.84 *** | 0.56 | **0.82 *** | **0.86 *** | −0.56 |
| K | 0.39 | 0.37 | 0.27 | 0.52 | −0.43 |
| Physical properties at end of growing season | | | | | |
| $K_{s\_1}$ | **−0.53 **** | −0.29 | −0.09 | **−0.42 *** | **0.41 *** |
| $\rho_{b\_1}$ | 0.33 | **0.63 **** | 0.28 | 0.35 | −0.14 |
| $\theta_{a\_1}$ | **−0.53 **** | −0.37 | −0.28 | **−0.51 *** | 0.23 |

[1] The smallest significant coefficient was 0.75 for variables measured after potting ($p < 0.05$; $n = 8$) and 0.40 for physical properties measured after one growing season ($p < 0.05$; $n = 24$). * Significant at $p < 0.05$, ** Significant at $p < 0.01$. [2] DW = dry weight, H = seedling height, D = root collar diameter and H/D = height–diameter ratio. $K_s$ = saturated hydraulic conductivity, $\rho_b$ = bulk density, $\theta_t$ = total porosity, $\theta_a$ = air-filled porosity at container capacity, MWD = mean weight diameter, <0.5 mm = fraction of peat particles with a diameter < 0.5 mm, 1–4 mm = proportion of peat particles with a diameter between 1 and 4 mm, EAW = easily available water content, AW = available water content, $\gamma$ = pore effectiveness coefficient, $D_s D_0^{-1}$ = relative gas diffusivity, EC = electrical conductivity and $CEC_{eff}$ = effective cation exchange capacity.

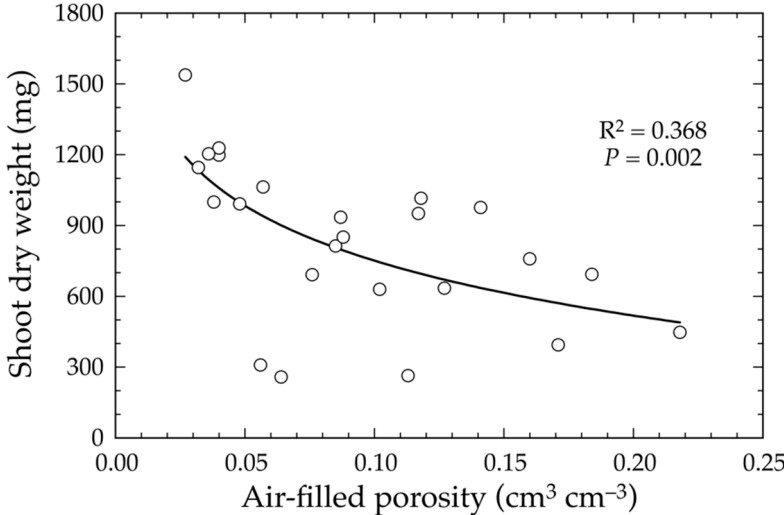

**Figure 5.** Relationship between shoot dry weight of white spruce seedlings (1+0) and the air-filled porosity at container capacity of substrates at the end of the growing season ($n = 24$).

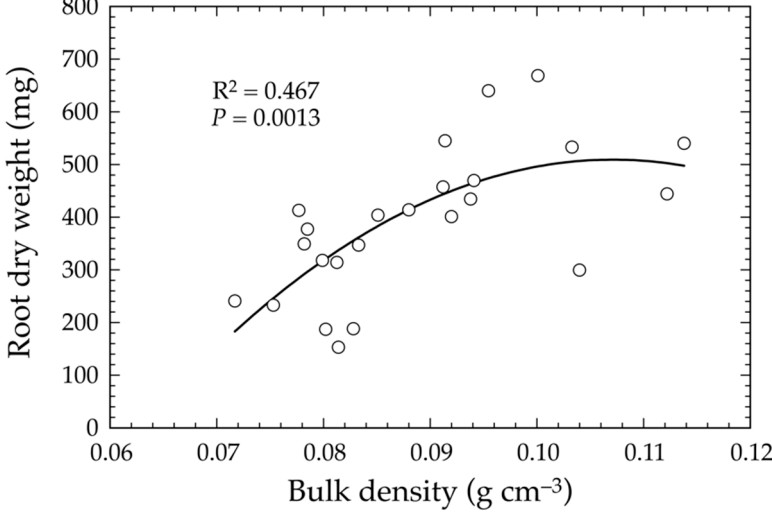

**Figure 6.** Relationship between root dry weight of white spruce seedlings (1+0) and bulk density of substrates at the end of the growing season ($n = 24$).

## 4. Discussion

　　Significant differences were observed between the physical properties of the substrates from the eight forest nurseries, despite the similarity of the different components of the substrates used to produce forest seedlings in Quebec [39]. The exception was the substrate from nursery 5, which contained a little more than 45% of particles larger than 1.7 mm (Figure 3). None of the substrates had a coarse (>2 mm)-to-fine particle ratio greater than 50%, as proposed as an acceptable standard for container forest seedling production [40]. It has been shown that the growth of white spruce seedlings produced in greenhouses was greater during the first year of production when the substrate was composed of fine particles (<1.3 mm) [9]. However, no significant relationship could be observed between MWD and growth variables during our study.

　　The $\theta_a$ of the substrates at container capacity measured just after potting varied between 0.04 and 0.17 $cm^3$ $cm^{-3}$ (Table 2) and was similar to $\theta_a$ values reported in the literature for this type of substrate [19,41,42]. Except for air-filled porosity observed in the substrate from nursery 1, these values were near the lowest limit for acceptable gas diffusion (i.e., $\theta_a$~0.10 $cm^3$ $cm^{-3}$) [43] as well as adequate root growth and respiration (i.e., $\theta_a$~0.10–0.15 $cm^3$ $cm^{-3}$) [44] and should likely have limited plant growth under a wet irrigation management. The significant negative correlations observed between $\theta_a$ and $K_s$ measured after one growing season and the growth variables instead indicated that low aeration of the substrate did not limit seedling growth, but instead suggested that too much aeration decreases shoot growth (Table 4, Figure 5). This would also be consistent with water stress affecting seedling growth. Indeed, these results are very similar to those obtained in a companion study for one-year growth (year 2008 in [20]). For aeration, this could be explained by the low need of conifers for aeration [45], especially during the first growing season, when the cavity volume is not very limiting for root growth. Moreover, for the correlation, $\theta_a$ was measured at container capacity ($\psi$ = −0.6 kPa), a matric potential that is rarely maintained for long periods after irrigation [42] as the irrigation threshold was most likely less than −5 kPa based on the recommended threshold of 0.30–0.45 $cm^3$ $cm^{-3}$ substrate water content to initiate irrigation.

　　Water stress appeared most likely to explain these negative correlations for the first year of growth, as previously reported [20]. Indeed, the irrigation system used in forest nurseries tends to maintain a volumetric water content between 0.30 and 0.45 $cm^3$ $cm^{-3}$. The desorption curves presented in Figure 4 well describe the reduction of water content in substrates and its replacement by air under a matric potential of −0.6 kPa, and indicate threshold water potentials to trigger irrigation far below −5 kPa and even −10 kPa for most substrates, threshold values below which growth limitations were reported for *Prunus × cystena* [46] and attributed to too-low values of saturated hydraulic conductivity. Results were also consistent with a decrease in shoot growth with decreasing water potential in our parallel study for the first year (2008 in [20]). Under a dry irrigation management, at water potentials below −2 kPa, $\theta_a$ was high (>0.15 $cm^3$ $cm^{-3}$) for most substrates from the eight nurseries. Values of $\theta_a$ at $\psi$ = −0.6 kPa (see Figure 4) were then likely uncommon in substrates during the growing season under forest nursery conditions, which greatly facilitated the presence of air-filled pores [6,7].

　　Aeration limitations in substrates also appeared very unlikely based on relative gas diffusivity measurements, even if wet conditions may have prevailed occasionally in the container between irrigation events. $D_s$ $D_0^{-1}$ values obtained in this study for peat substrates used in eight forest nurseries (0.0041–0.0171 $m^2$ $s^{-1}$ $m^{-2}$ s) were similar to those observed by Boudreault et al. [20] ($D_s$ $D_0^{-1}$ = 0.0026 to 0.0105 $m^2$ $s^{-1}$ $m^{-2}$ s), Caron et al. ([47]; $D_s$ $D_0^{-1}$ = 0.0026 to 0.0105 $m^2$ $s^{-1}$ $m^{-2}$ s) and Caron and Nkongolo ([22]; $D_s$ $D_0^{-1}$ = 0.0067 to 0.0230 $m^2$ $s^{-1}$ $m^{-2}$ s) using the same method. Allaire et al. [18] reported $D_s$ $D_0^{-1}$ values that ranged between 0.006 and 0.018 $m^2$ $s^{-1}$ $m^{-2}$ s for horticultural substrates. Pore effectiveness and $D_s$ $D_0^{-1}$ measured at container capacity did not show any significant correlation with seedling growth variables (Table 4) contrary to several studies where positive linear relationships were observed between these variables

for a similar $D_s D_0^{-1}$ range in peat and bark substrates [18,48,49]. For instance, Allaire et al. [18] established a relative gas diffusivity threshold of 0.015 $m^2 s^{-1} m^{-2}$ s to promote the growth of *Prunus × cistena* in 5 L (21 cm high × 18 cm diameter) containers. Meanwhile, Boudreault et al. [20] suggested, based on modeled $O_2$ profiles in root plugs (310 $cm^3$ cavities; 12 cm high × 6.5 cm diameter with ~5 $cm^2$ hole at bottom for drainage and air exchange) and observed $D_s D_0^{-1}$–root growth relationship for the same tree seedling species (see Figure 5 in [20]), that nursery substrates should have a relative gas diffusivity greater than ~0.004 $m^2 s^{-1} m^{-2}$ s when matric potential is maintained within the 0 to −5 kPa range. This apparent discrepancy between relative gas diffusivity minimal values may be due to the fact that the seedlings from forest nurseries were grown in cavities of reduced height and radius compared to the pots commonly used in horticultural nurseries (1 L and more). Unlike horticultural pots, the cavities of containers used by forest nurseries had a greater surface area available for gas exchange relative to volume. This surface is made up of the upper part (top) of the root plug, but also of the contour of the cavity, where gas exchange capacity is accentuated by root water uptake during substrate drying (see Boudreault et al. [20]). Moreover, the bottom of the root plug is directly exposed to air, as opposed to the 5 L containers in Allaire et al. [18], which had five small drainage holes at the bottom of the container not directly exposed to air. Hence, although the $D_s D_0^{-1}$ values measured here were typically below the 0.015 $m^2 s^{-1} m^{-2}$ s threshold, they were higher than the 0.004 $m^2 s^{-1} m^{-2}$ threshold obtained by Boudreault et al. [20]), and therefore the white spruce seedling growth was unlikely limited by aeration.

The level of peat humification, reduction in particle size and compaction of the substrate at potting can increase bulk density, which also generally increases during the growing season [9,11]. A significant positive correlation was observed between bulk density and root dry mass (Table 4; Figure 6). Low bulk density could affect plant water supply, resulting in lower total plant mass [20]. Unsaturated hydraulic conductivity is also low in peat substrates under dry conditions [46,50,51]. On the other hand, large particle size combined with low bulk density can reduce unsaturated hydraulic conductivity, which could limit water transfer for substrates under dry irrigation management, as suggested by our data and previous work for the same species [20].

Total porosity values for sampled substrates were similar to those reported in the literature for blond and brown fibric peat [11,16,52–54]. Moreover, substrates from the eight forest nurseries showed desorption curves similar to those observed in other studies [10,54]. Overall, the available and easily available water content varied little among substrates (Table 2). This is not surprising, as the volumetric water content at −5 and −10 kPa were similar, except for the substrate from nursery 1, which had a generally higher water content (Figure 4). Bernier and Gonzalez [9] observed a positive correlation between root dry mass and easily available water. However, this relationship was not observed in the present study, which may be due to irrigation management and environmental conditions between nurseries.

Given the importance of bulk density when packing growing media in seedling trays at the nursery operation scale, the shape of the curve in Figure 6 is of practical interest and should be analyzed cautiously. First, the results suggest a plateau around 0.10 to 0.11 $g cm^{-3}$, with a possible decrease beyond 0.11 $g cm^{-3}$. Higher values may possibly lead to growth decreases, as reported by Boudreault et al. [20] for their second-year (2009) data, when seedlings were exposed to outdoor conditions. Further, the conclusion applies to the first year only, as with time the substrate will significantly evolve and become more compacted. Seedling responses will also be altered in the second year, when plants will be brought into outdoor conditions for their second-year growth, which may expose the substrate to prolonged wet conditions [20]. Boudreault et al. [20] consistently reported results similar to those above for the first year, but observed important aeration limitations when substrates were more compacted in the second year.

Fertilization applied during the growing season influenced seedling growth and development, as demonstrated by correlations between substrate fertility ($N_{total}$ and P

concentrations) and shoot dry mass, height and root-collar diameter (Table 4). The amount of mineral nutrients needed in each forest nursery during the growing season (Table 1) is calculated based on growth standards to be achieved (height, root and shoot dry masses, etc.) in order to optimize the concentration of nutrients in the substrate and in shoot and root tissues [24]. Hence, the different fertilization regimes used by nursery growers may have masked some of the relationships between the physicochemical properties of the substrates and the growth variables of white spruce seedlings. The concentrations of mineral nutrients in shoots were, however, within recommended ranges for the growth and physiological processes (net photosynthesis, etc.) of white spruce seedlings in all nurseries (for instance, foliar nitrogen was 2.4% to 3%; Table 3) [55–57]. Before the delivery of seedlings to reforestation sites and payment to forest nurseries in Quebec, seedlings are evaluated and must meet 27 morphophysiological quality standards, including leaf nitrogen concentration [3]. To this end, nurseries must respect the optimal thresholds of the main cultural practices (peat substrate, irrigation, fertilization, short-day treatment, etc.) to achieve these 27 quality standards and reduce seedling losses and production costs [4,58,59]. Another factor to be considered when examining relations between the physical properties of substrates and conifer seedling biomass is root growth throughout the growing season. As roots develop, they occupy the pore space of the growing medium, consume oxygen and affect gas diffusion in the rhizosphere, which may also impact root and shoot growth [60–62].

## 5. Conclusions

This characterization of the physicochemical properties of substrates from eight forest nurseries highlighted a range of quality in local peat. While substrate chemical properties were similar, their physical properties varied significantly. The low intrinsic aeration of substrates did not limit the root and shoot growth of white spruce seedlings (1+0). Conversely, seedling growth was lower in substrates with high air-filled porosity. Our findings suggest that the air-filled porosity at container capacity of forest nursery substrates should be between 0.03 and 0.10 $cm^3$ $cm^{-3}$ to improve the root and shoot growth of white spruce seedlings (1+0). Moreover, a substrate bulk density between 0.07 and 0.115 $g$ $cm^{-3}$ had a positive effect on root biomass. This is likely due to irrigation management resulting in drier substrate conditions during this first year of growth under a polyethylene tunnel, where controlled irrigation maintains relatively low water content and adequate air-filled porosity. However, a few forest nursery substrates had low air-filled porosity values at container capacity and low pore effectiveness coefficients, which could hinder root growth and development during the second growing season when seedlings are cultivated outdoors and thus exposed to prevailing weather conditions (i.e., rainfall). This hypothesis should be tested by further study of their second year of growth.

**Author Contributions:** Conceptualization, S.P., J.C., M.S.L. and S.B.; methodology, S.P., J.C., S.B. and M.S.L.; software, S.B.; validation, S.B., S.P., J.C. and M.S.L.; formal analysis, S.B.; investigation, S.B., S.P., J.C. and M.S.L.; resources, S.P.; data curation, S.B.; writing—original draft preparation, S.B., S.P., M.S.L. and J.C.; writing—review and editing, S.B., S.P., M.S.L. and J.C.; visualization, S.B.; supervision, S.P., J.C. and M.S.L.; project administration, S.P.; funding acquisition, S.P. All authors have read and agreed to the published version of the manuscript.

**Funding:** Funding for this project was provided by the Fonds de recherche du Québec—Nature et technologies (Partnerships Research Program on Forest Management and Environment III: grant n° 2008-FT-124361), awarded to S. Pepin.

**Data Availability Statement:** Not applicable.

**Acknowledgments:** The authors would like to thank the forest nurseries that provided us with potting substrates and tree seedlings, as well as Jean-François Bernier, Anne-Isabelle Bonifassi, Marie-Ève Giroux and Andrée-Ann Prince for their involvement in this project. We would also like to thank Carole Boily (Laval University), Bertrand Fecteau and Luc Godin (pépinière Pampev Inc.) and the

staff of the Organic and Inorganic Chemistry Laboratory of the Forest Research Branch of the Quebec Ministry of Natural Resources and Forests for their participation in this study.

**Conflicts of Interest:** The authors declare no conflict of interest.

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
