# Peer review of "Comparison of Hydraulic and Aeration Properties of Peat Substrates Used to Produce Containerized White Spruce Seedlings (1+0) in Forest Nurseries"

_forests, doi:10.3390/f14040858_

Round 1

Reviewer 1 Report

The manuscript ‘Comparison of hydraulic and aeration properties of peat substrates used to produce containerized white spruce seedlings (1+0) in forest nurseries’ is well organized and written.

I have only a few considerations to add.

Keywords:

‘peat substrate’ it's already in the title, please replace

Introduction:

Line 27 - indicate here for readers where Quebec is located

Lines 65-67 – this sentence is not clear

Lines 70-74 – please, rewrite

Table 2: no letters are given for EAW, ϒ, DsD0-1, if statistically not significant please add specification

Discussion:

Line 399-401: it is not clear what Authors would like to express

Author Response

Reviewer 1:

The manuscript ‘Comparison of hydraulic and aeration properties of peat substrates used to produce containerized white spruce seedlings (1+0) in forest nurseries’ is well organized and written. We thank you for your comment.

I have only a few considerations to add.

  • Keywords: ‘peat substrate’ it's already in the title, please replace: replaced by growing medium
  • Line 27 – indicate here for readers where Quebec is located: Canada was added.
  • Lines 65-67 – this sentence is not clear: the sentence was replaced by “Moreover, the height of the saturated zoneat the bottom of the root plug after irrigation or rainfall events increases in fine-textured peat substrates, which may further limit gas exchange in the rhizosphere [15]”.
  • Lines 70-74 – please, rewrite: the two sentences were replaced by “However, gas exchange between substrates and ambient air also depends on pores connection in the growing medium and consequently, high θa values are not always indicative of adequate substrate aeration. The relative diffusivity of gases (DD0–1; estimated from measurements of θa, pore tortuosity and saturated hydraulic conductivity) has been shown to be a better parameter than air-filled porosity to investigate how substrate aeration properties affect the development and growth of plant species [21, 22].”
  • Table 2: no letters are given for EAW, ϒ, DsD0-1, if statistically not significant please add specification: letters ‘a’ have been added to all EAW, g and DsD0–1 values in Table 2.
  • Discussion: Line 399-401: it is not clear what Authors would like to express: the sentence was rewritten as followed: “Aeration limitations in substrates also appeared very unlikely based on relative gas diffusivity measurements, even if wet conditions may have prevailed occasionally in the container between irrigation events.”

Reviewer 2 Report

The work presented for review entitled “Comparison of hydraulic and aeration properties of peat sub-2 strates used to produce containerized white spruce seedlings 3 (1+0) in forest nurseries”, is an example of a very interesting work.

Introduction

The introduction of the work is written very well, summarizing the most important known information on the impact of the basic physical parameters of the substrate on the growth of seedlings.

Materials and methods

1. In the Materials and Methods chapter, information on the height and width of the cell of the IPL 25-310: 25 container should be included, because it is important for the measurement with the TDR sensor, whose working elements are 12 cm long, and the cell width will be important in relation to to the width of the TDR sensor. This information is hard to find on the Internet.

2. Important information to be completed is the method of filling the containers with the substrate. Was it identical in all nurseries? Was it carried out using a vibrating table and with what operating parameters of this table?. This information is important for evaluating and comparing the physical parameters of substrates between nurseries. Do the obtained differences result from the substrate itself or from the method of filling the substrate into the containers?.

3. The obtained parameters, especially with regard to air capacity, are very low, on the borderline that can be tolerated by plants, how to explain such low values?

4. θt appears in the methodology, for which there is no explanation.

5. In the methodology chapter, I would suggest a photo, diagram or illustrative drawing for the measurement method described in paragraphs 152-160. It is not possible to access all literature and the description is difficult to read.

Description of the results obtained, discussion, and conclusions are well written.

I encourage you to familiarize yourself with publications very related to the subject matter, perhaps something will be of interest to you.

Kormanek, M., MaÅ‚ek, S., Banach, J. et al. Seasonal changes of perlite–peat substrate properties in seedlings grown in different sized container trays. New Forests 52, 271–283 (2021). https://doi.org/10.1007/s11056-020-09793-3

Author Response

Reviewer 2

The work presented for review entitled “Comparison of hydraulic and aeration properties of peat substrates used to produce containerized white spruce seedlings (1+0) in forest nurseries”, is an example of a very interesting work.

Introduction

The introduction of the work is written very well, summarizing the most important known information on the impact of the basic physical parameters of the substrate on the growth of seedlings. Thank you.

Materials and methods

  1. In the Materials and Methods chapter, information on the height and width of the cell of the IPL 25-310 container should be included, because it is important for the measurement with the TDR sensor, whose working elements are 12 cm long, and the cell width will be important in relation to the width of the TDR sensor. This information is hard to find on the Internet. We thank reviewer 2 for his comment as we noticed that the TDR probe length was not 12 cm but rather 10.5 cm long. More details on the TDR probe length and rod diameter and spacing are now provided on lines 154–155. Additionally, we included the cavity height and width of the IPL 25-310 containers on line 113.

  1. Important information to be completed is the method of filling the containers with the substrate. Was it identical in all nurseries? Was it carried out using a vibrating table and with what operating parameters of this table?. This information is important for evaluating and comparing the physical parameters of substrates between nurseries. Do the obtained differences result from the substrate itself or from the method of filling the substrate into the containers? Unfortunately, we do not know exactly all the details on the methods used by the different nurseries for filling their 25-310 containers and only one nursery manager mentioned a vibrating table without providing any operating parameters. Moreover, forest nurseries purchased their peat from different suppliers. This is why we advocate for a characterization of hydraulic and aeration properties of substrates after potting.

  1. The obtained parameters, especially with regard to air capacity, are very low, on the borderline that can be tolerated by plants, how to explain such low values?

The air capacity may indeed seem low. There are several reasons for this. First of all, air capacity was measured directly in the container at an applied water potential (-0.6 kPa) representative of the average suction observed at the mid-cavity height (6 cm). Standard methods in the literature usually report measurements at -1 kPa, a higher suction which will obviously end up in higher air-filled porosity values. Second, measurements were performed for several nurseries for which we had no control neither on the substrate origin nor on the potting procedure, even though they all used the same recommendations from the Ministry of Forests. This brings additional variability in the parameters, as observed in the results. Third, the measurements were performed straight after potting and after one growing season. Most of the results obtained after potting (except for nursery 1) are close or above a 10% level, which is usually appropriate given a container configuration with openings to provide enough aeration. Measurements taken after one year will obviously be affected by substrate setting, decomposition, and root growth, which will result in lower air-filled porosity values.

Despite being low, the air-filled porosities achieved in the container remained appropriate from what we see from seedling growth. Indeed, reaching a bulk density of about 0.10 g cm-3 is necessary to get top yields which will create a substrate with less air-filled porosity. Other results in the study confirm that aeration limitations did not appear to have influenced negatively this first year of growth. However, the reviewer is right with respect to possible aeration limitation at some points. Our data suggest that reaching higher bulk densities (above 0.10-0.11 g cm-3) may be detrimental to seedling growth, as highlighted in the discussion based on the shape of the curve relating plant productivity to substrate bulk density (Fig. 6).

  1. θt appears in the methodology, for which there is no explanation. θt is now defined on line 135 in the revised manuscript.

  1. In the methodology chapter, I would suggest a photo, diagram or illustrative drawing for the measurement method described in paragraphs 152-160. It is not possible to access all literature and the description is difficult to read. We added a schematic of the five-cavity container with TDR probes (now Fig. 2)

Description of the results obtained, discussion, and conclusions are well written. We thank referee 2 for this comment.

I encourage you to familiarize yourself with publications very related to the subject matter, perhaps something will be of interest to you. Thanks for pointing out this paper to us.

Kormanek, M., MaÅ‚ek, S., Banach, J. et al. Seasonal changes of perlite–peat substrate properties in seedlings grown in different sized container trays. New Forests 52, 271–283 (2021). https://doi.org/10.1007/s11056-020-09793-3

Round 2

Reviewer 1 Report

The required revisions and changes have been incorporated into the new version and therefore the manuscript is eligible for publication.

Reviewer 2 Report

The corrections and explanation of the Authors are enough for me